# Anemia in Dogs with Acute Kidney Injury

**DOI:** 10.3390/vetsci11050212

**Published:** 2024-05-13

**Authors:** Ilaria Lippi, Francesca Perondi, Giulia Ghiselli, Sara Santini, Verena Habermaass, Veronica Marchetti

**Affiliations:** Dipartimento di Scienze Veterinarie, Università di Pisa, 56126 Pisa, Italy; f.perondi87@gmail.com (F.P.); g.ghiselli2@studenti.unipi.it (G.G.); s.santini13@studenti.unipi.it (S.S.); verena.habermaass@phd.unipi.it (V.H.); veronica.marchetti@unipi.it (V.M.)

**Keywords:** anemia, AKI, dog, IRIS grades

## Abstract

**Simple Summary:**

Anemia is universally recognized as a common finding of both human and veterinary chronic kidney disease, but its role in acute kidney injury is still debated. The retrospective evaluation of medical records of AKI dogs showed that anemia was a common disorder, which was found in 72% of the study population. The majority of dogs with AKI showed mild to moderate, normochromic, normocytic, and non-regenerative anemia. The frequency of anemia was elevated for all the AKI etiologies, with increasing frequency with the progression of the IRIS grade. Anemia should be considered a common disorder in AKI dogs and should be promptly addressed and treated.

**Abstract:**

Anemia is a well-known complication in CKD dogs, but its frequency in AKI dogs has been poorly investigated. The aim of the present study was to retrospectively evaluate frequency, degree of severity, and regeneration rate of anemia in relation to IRIS grade, etiology, therapy, and outcome. Medical records of dogs (2017–2023) with historical, laboratory, and ultrasound findings consistent with AKI were retrospectively reviewed. According to etiology, AKI was classified as ischemic/inflammatory (IS), infectious (INF), nephrotoxic (NEP), obstructive (OBS), and unknown (UK). AKI dogs were also classified according to therapeutical management (medical vs. hemodialysis), survival to discharge (survivors vs. non-survivors). Anemia was defined as HCT < 37% and classified as mild (HCT 30–37%), moderate (HCT 20–29%), severe (13–19%), or very severe (<13%). Anemia was classified as microcytic (MCV < 61 fL), normocytic (61 and 73 fL), and macrocytic (>73 fL). Anemia was considered hypochromic (MCHC< 32 g/dL), normochromic (32 and 38 g/dL), and hyperchromic (>38 g/dL). Regeneration rate was considered absent (RET ≤ 60,000/μL), mild 61,000–150,000/μL), and moderate (>150,000/μL). A total of 120 AKI dogs were included in the study, and anemia was found in 86/120 dogs (72%). The severity of anemia was mild in 32/86 dogs (37%), moderate in 40/86 dogs (47%), severe in 11/86 dogs (13%), and very severe in 3/86 (3%). Anemia was normochromic in 71/86 dogs (83%), hyperchromic in 12/86 dogs (14%), and hypochromic in 3/86 dogs (3%). Normocytic anemia was present in 56/86 dogs (65%), microcytic anemia in 27/86 dogs (31%), and macrocytic anemia in 3/86 dogs (4%). Non-regenerative anemia was found in 76/86 dogs (88%). The frequency of anemia increased significantly (*p* < 0.0001) with the progression of IRIS grade, although no significant difference in the severity of anemia was found among the IRIS grades. The frequency of non-regenerative forms of anemia was significantly higher than regenerative forms (*p* < 0.0001) in all IRIS grades. In our population of AKI dogs, anemia was a very frequent finding, in agreement with current findings in human nephrology.

## 1. Introduction

Acute kidney injury (AKI) is a complex clinical syndrome, characterized by an abrupt reduction of glomerular filtration rate (GFR) and subsequent uremia [1]. In veterinary medicine, the most common causes of AKI are represented by hemodynamic disorders (e.g., hypotension or hypovolemia), infections (e.g., Leptospirosis, or pyelonephritis), nephrotoxic agents (e.g., ethylene glycol, lilies, grape, NSAIDs), obstruction, and/or rupture of the urinary tract. However, a definitive diagnosis of AKI is not always available, and the cause of AKI may remain unknown. AKI is associated with a mortality rate of approximately 46%, with variable rates according to etiology, treatment (medical vs. hemodialysis), and occurrence of complications [2]. In human patients with AKI, anemia was found to be a very frequent disorder, affecting approximately 90% of the population [3]. AKI may contribute to anemia through different mechanisms, such as reduced life span of RBCs, increased risk of bleeding, and reduced production of erythropoietin (EPO), although the association between anemia and AKI progression is controversial [3,4]. Impaired EPO production is postulated to occur secondary to the injury of interstitial fibroblasts. Damaged fibroblast may cause a significant reduction in the local production of EPO, thus contributing to the development of anemia. In human patients submitted to non-cardiac surgery, a higher degree or longer duration of AKI was associated with more severe forms of anemia during the first year after surgery. In this population, the post-surgery frequency of anemia was significantly higher in patients with AKI, compared to non-AKI [5]. A retrospective observational study conducted on human patients undergoing cardiopulmonary bypass showed a 16% increased risk of postoperative AKI for every g/dL reduction in hemoglobin [6]. Another study in patients with cardiopulmonary bypass showed that a hematocrit < 21% increased the risk for AKI by 2.34 times, compared to patients with a hematocrit between 21% and 25% [7]. Hemodilution may reduce the oxygen-carrying capacity of the blood, causing an impaired delivery of oxygen to the renal medulla [6]. The association between pre-surgery anemia and increased risk of AKI was also reported by a large retrospective cohort study, which investigated possible risk factors for AKI in human patients undergoing different kinds of surgery. In this population, the presence of pre-surgery anemia significantly increased the risk of post-surgery AKI and the need for dialysis [8].

In CKD dogs, anemia is a well-known disorder, whose frequency and severity increase with the progression of the IRIS stage [9,10]. In these dogs, anemia was found to be mostly non-regenerative, with a strong association between low reticulocyte count and progression of the IRIS stage, due to several mechanisms, such as decreased blood flow and oxygen delivery, increased oxidative stress, and stimulation of renal fibrosis [10]. In AKI dogs of different etiologies, anemia was reported in a significant percentage of the population (32%), with higher frequency in non-survivors (42%), compared to survivors (29%) [1].

The present study is a retrospective survey designed to evaluate the frequency of anemia in AKI dogs, its degree of severity, and regeneration rate in relation to IRIS grade, etiology, therapy, and outcome.

## 2. Materials and Methods

### 2.1. Case Selection

The medical records of dogs diagnosed with AKI, referred from the Veterinary Teaching Hospital “Mario Modenato” of the Department of Veterinary Sciences of the University of Pisa between January 2017 and January 2023, were retrospectively reviewed.

Dogs with historical, laboratory, and ultrasound findings consistent with AKI were included in the study. AKI was diagnosed based on the acute onset of clinical signs consistent with AKI (e.g., vomiting, loss of appetite, oliguria/anuria, polyuria), and on the guidelines for the diagnosis and grading of AKI of the International Renal Interest Society (IRIS) [11]. Dogs were classified in IRIS AKI grades, on the basis of the serum creatinine they showed at hospital admission. AKI dogs were included in the study if they had the availability of complete blood count (CBC), serum creatinine, and urea at hospital admission. Dogs were excluded from the study if one or more of the following conditions were present: (1) historical, laboratory, and/or ultrasonographic findings consistent with chronic kidney disease (CKD) [12]; (2) use of alpha-darbepoetin or red blood cell transfusion prior to the presentation; (3) rechecks of the same patients (since the same dog was included in the case selection only once); (4) severe comorbidities, which might affect CBC. Included dogs were classified according to the IRIS grading of AKI, as follows: IRIS grade 1 (serum creatinine < 1.6 mg/dL), IRIS grade 2 (serum creatinine between 1.7 mg/dL and 2.5 mg/dL), IRIS grade 3 (serum creatinine between 2.6 mg/dL and 5 mg/dL), IRIS grade 4 (serum creatinine between 5.1 mg/dL and 10 mg/dL), and IRIS grade 5 (serum creatinine > 10 mg/dL). AKI dogs were sub-graded according to urinary production (oliguria vs. no oliguria) and need for renal replacement therapy (RRT). Oliguria was defined as a urinary production < 1 mL/kg/h over 6 h [13].

### 2.2. Etiology of AKI

According to etiology, AKI was classified as ischemic/inflammatory (IS), infectious INF), nephrotoxic (NEP), obstructive (OBS), and unknown (UK). The ischemic/inflammatory group included dogs for which AKI developed as a consequence of one of the following conditions: acute pancreatitis, heatstroke, pyometra, or snake envenomation. Acute pancreatitis was diagnosed, on the basis of clinical signs (abdominal pain, vomiting, anorexia), abdominal ultrasound (hypoechoic areas within the pancreatic parenchyma, hyperechoic mesenteric areas surrounding the pancreas, enlarged pancreas with irregular profile, abdominal effusion), and abnormal IDEXX SNAP cPL test. The infection group included dogs in which AKI was caused by Leptospirosis, bacterial pyelonephritis, or vector-borne diseases (Babesia, Ehrlichia, Leishmania). The nephrotoxic group included dogs for which AKI developed as a consequence of the ingestion of a nephrotoxic substance (e.g., grape, raisin, ethylene glycol). The obstructive group included dogs in which AKI was caused by obstructive kidney disease (uroliths, neoplasia, iatrogenic ureteral injury). The unknown group included dogs for which the etiology of AKI was unknown.

### 2.3. Therapeutical Management and Outcome

According to therapeutical management, AKI dogs were divided into two groups: medical (M), and hemodialysis (HD). The medical management group included dogs with less severe forms of azotemia (e.g., serum creatinine < 8 mg/dL), non-oliguric and non-hyperkalemic AKI, and dogs with severe forms of azotemia (serum creatinine > 8 mg/dL), which could not be managed by hemodialysis due to financial constraints. The hemodialysis group included dogs with more severe forms of azotemia, oliguric and/or hyperkalemic dogs, and AKI dogs with fluid overload. According to the outcome, AKI dogs were divided into survivors (S) and non-survivors (NS). Dogs that survived hospital discharge were considered survivors, while dogs that died or were euthanized during hospitalization were considered non-survivors.

### 2.4. Hematological and Biochemical Parameters at Hospital Admission

Complete blood counts (CBCs) were performed using a laser cell counter (Procyte DX, IDEXX Laboratories, Westbrook, ME, USA), and a blood smear stained with May–Grünwald Giemsa (Aerospray Wescor, Delcon, Milan, Italy) was microscopically examined by experienced and trained clinical pathologists. For CBCs the following parameters were evaluated: red blood cell count (RBC), hematocrit (HCT), hemoglobin (HGB), mean corpuscular volume (MCV), mean corpuscular hemoglobin (MCH), mean corpuscular hemoglobin concentration (MCHC), red blood cell distribution width (RDW), and absolute reticulocytes count (RET). AKI dogs were considered anemic in the case of HCT < 37% [14]. According to HCT, anemia was classified as mild (HCT 30–37%), moderate (HCT 20–29%), severe (HCT 13–19%), or very severe (HCT < 13%) [14]. Anemia was considered microcytic for MCV < 61 fL, normocytic for MCV between 61 and 73 fL, and macrocytic for MCV > 73 fL. Anemia was considered hypochromic for MCHC < 32 g/dL, normochromic for MCHC between 32 and 38 g/dL, and hyperchromic for MCHC > 38 g/dL. According to the absolute reticulocyte count, regeneration rate was considered absent for RET ≤ 60,000/μL, mild for RET ranging between 61,000 and 150,000/μL, and moderate for RET > 150,000/μL [14].

Serum creatinine, urea, total calcium, phosphate, sodium, and potassium were assessed by using liquid chemistry (instrument SAT450 or Liasys, with dedicated reagent kits, Analyzer Medical System—AMS, Rome, Italy).

### 2.5. Statistical Analysis

Continuous variables were tested for normality through the Kolmogorov–Smirnov test.

Median values of non-parametric variables (RBC, MCV, MCH, MCHC, RDW, %Ret, RET, serum creatinine, sodium, potassium) were compared by Mann–Whitney test when the comparison was between dogs treated with medical management and hemodialysis and between survivors and non-survivors. Kruskal–Wallis test was used to compare median values of non-parametric variables among dogs at different IRIS grades and among dogs with different etiologies of AKI. Mean values of parametric variables (HCT, HGB, serum urea, total calcium, phosphate) were compared by unpaired *t*-test when the comparison was between dogs treated with medical management and hemodialysis, and between survivors and non-survivors. One-way ANOVA was used to compare mean values of parametric variables among dogs at different IRIS grades and among dogs with different etiologies.

Chi-squared test of independence was used to compare the frequency of anemia, the severity of anemia, and the regeneration rate in AKI dogs at different IRIS grades, etiologies, therapeutical management, and outcomes.

Statistical analysis was performed using commercial statistical software (GraphPad Prism9 for macOS, GraphPad Software Inc., San Diego, CA, USA), and data were considered statistically significant for *p*-values < 0.05.

## 3. Results

The retrospective analysis of the electronic database of the Veterinary Teaching Hospital “Mario Modenato” of the Department of Veterinary Sciences of the University of Pisa found 220 dogs with AKI, between January 2017 and January 2023. Of the initially selected 220 dogs, 62 dogs were excluded due to pre-existing CKD, 23 dogs were excluded as they already received blood transfusion or darbepoetin supplementation, and 15 dogs were excluded due to missing laboratory data. A total of 120 dogs with AKI were included in the study. Enrolled dogs were distributed in intact females (n = 23), sterilized females (n = 25), intact males (n = 69), neutered males (n = 3), of different breed: mix-breed (n = 35); Labrador (n = 11); Border Collie (n = 6); Jack Russell, Springer Spaniel, Pointer (n = 5); Beagle, Australian Shepherd, English Setter, German Shepherd (n = 4); Boxer, Dachshund, Golden Retriever (n = 3); Epagneul Breton, Italian Spinone, Weimaraner, Mastiff (n = 2); Akita, Poodle, Bull Mastiff, Bull Terrier, French Bulldog, Bloodhound, Siberian Husky, American Staffordshire, Cocker Spaniel, Flat Coated Retriever, Lagotto Romagnolo, Czechoslovakian Wolfdog, Maltese, Great Pyrenees, Central Asian Shepherd, Pitt Bull, Rottweiler, Shih-Tzu, Yorkshire (n = 1). Median age was 4.1 years (1 month-15 years). According to the IRIS grading system, AKI dogs were distributed in: IRIS grade 2 (n = 10); IRIS grade 3 (n = 22); IRIS grade 4 (n = 48); and IRIS grade 5 (n = 40). The mean and median values of hemato-biochemistry parameters of AKI dogs at hospital admission are reported in Table 1. According to etiology, AKI dogs were distributed in: ischemic/inflammatory (17/120; 14%), infective (36/120; 30%), nephrotoxic (14/120; 12%), obstructive (5/120; 4%), and unknown (48/120; 40%). According to urine production, oliguria was found in 61/120 dogs (51%), while normal urine production was found in 59/120 dogs (49%). Medical management was performed in 50/120 dogs (42%), while hemodialysis was performed in 70/120 dogs (58%). According to the outcome, 66/120 dogs (55%) survived, while 54/120 (45%) dogs died.

Anemia was found in 86/120 dogs (72%). According to severity, 37% of the dogs (32/86) showed mild anemia, 47% (40/86) showed moderate anemia, 13% (11/86) showed severe anemia, and 3% (3/86) showed very severe anemia. Moreover, 83% of anemic dogs (71/86) showed normochromic anemia, while 14% (12/86) and 3% (3/86) of dogs showed hyperchromic and hypochromic anemia, respectively. According to MCV, 65% of dogs showed normocytic anemia (56/86), 31% microcytic anemia (27/86), and 4% macrocytic anemia (3/86). Non-regenerative anemia was found in 88% of anemic dogs (76/86), while regenerative anemia was found in 12% (10/86) of the dogs. According to the grade of AKI, the frequency of anemia was 40% (4/10) in IRIS grade 2, 64% (14/22) in IRIS grade 3, 83% (40/48) in IRIS grade 4, and 70% (28/40) in IRIS grade 5. The frequency of anemia increased significantly (*p* < 0.0001) with the progression of the IRIS grade, while no significant difference in the severity of anemia was noticed (Table 2). The frequency of non-regenerative forms of anemia was significantly higher than regenerative forms (*p* < 0.0001) in all IRIS grades. The frequency of mild to moderate degree of anemia, compared to severe to very severe degree of anemia did not differ significantly among the IRIS grades of AKI (*p* = 0.163).

The frequency of anemic dogs was significantly higher (*p* < 0.0001) than non-anemic dogs for all the causes of AKI, with the exception of toxic causes.

## 4. Discussion

In our population of AKI dogs, anemia was a very frequent finding, with an overall frequency of 72%, mostly normochromic (83%) and normocytic (65%) and characterized by mild (37%) to moderate (47%) severity and poor regeneration. This was a very interesting finding as in veterinary medicine, this kind of anemia is usually associated with chronic kidney disease (CKD), rather than AKI. In CKD dogs, anemia is mainly caused by the reduction of the production of erythropoietin (EPO), together with other mechanisms, such as chronic inflammation, disorders of iron metabolism, blood loss, and reduction in erythrocyte survival [10]. Although serum concentration of EPO was not available for the dogs of our study, it is plausible that deficiency of EPO may play a role in the pathogenesis of anemia, as the elevated frequency of no-regeneration. A previous study, which we conducted in CKD dogs, showed that anemia affected 63% of the canine population, with a significant increase in its frequency with the progression of the IRIS stage [10]. The elevated frequency of anemia in our population of AKI dogs was in line with human medicine, in which anemia is considered a common disorder [15]. In human medicine, a causal relationship between AKI and anemia was identified, not only because AKI is a timely antecedent to anemia, but also because higher grades of AKI, or longer duration of the disease are usually associated with more severe forms of anemia at three and six months. The association between AKI and anemia is present also after correction for potential confounding factors, and its occurrence is independent of variations in the glomerular filtration rate (GFR) [5]. Although the relationship between AKI and anemia in dogs was not specifically considered, a recent study by Rimer D. and colleagues investigating clinicopathologic findings of AKI dogs reported a frequency of anemia of approximately 32% [1]. The higher frequency of anemia in our study group compared to Rimer D. and colleagues may be related to several factors. Although both populations were constituted by AKI dogs, they showed differences in relation to geographic location, etiology, and severity of AKI grade. As the main investigation of our study was to assess the frequency of anemia in AKI dogs at hospital admission, we opted not to include AKI dogs that had already received darbepoetin, and/or blood transfusion. In the study by Rimer D. and colleagues, the use of darbepoetin or blood transfusion was not considered an exclusion criterion, so it is possible that some dogs in their study did not show anemia as a consequence of the use of these therapies. Another possible explanation for the higher frequency of anemia in our study may be the different distribution of dogs according to etiologies. In our population, AKI secondary to infection was found in 30% of the dogs, compared to the 8% in the study by Rimer D. and colleagues [1]. The association between anemia and infection has been reported, as a consequence of multiple factors. During infection, the hepatic synthesis of hepcidine is stimulated by several inflammatory cytokines, such as IL-6, IL-1, and TNF_α_. As serum levels of hepcidine increase, iron availability reduces, despite normal to elevated iron stores. The main consequence of the reduction of iron availability is a defective response of erythropoiesis to endogenous erythropoietin, and to erythropoietin stimulating agents [16,17]. Another possible cause of anemia during infection is represented by hemolysis. Although some bacteria and viruses may produce hemolysin in cultures, this is a rare cause of hemolysis in vivo. However, some bacteria can be directly responsible for hemolysis of RBCs by the release of toxins, which interfere with the erythrocyte membrane, as reported in infections due to *Clostridium perfringens*. Infection of erythrocytes (as seen in Babesiosis) can lead to a loss of plasticity of the membrane of RBCs and to the activation of their surface adhesive molecules, thus promoting microvascular hyper-viscosity and damage to RBCs. The fragility of RBCs might also be responsible for the 14% of anemic dogs with elevated MCHC due to the release of hemoglobin. Infections may also be associated with significant blood loss due to platelet dysfunction and disseminated intravascular coagulation (DIC). Severe forms of acute infections can cause DIC, by promoting disorders of the coagulation pathway. During bacterial infections, lipopolysaccharides (LPS) may be produced by bacteria and cause direct endothelial injury, such as microangiopathic hemolysis. Besides this mechanism, the increase in some inflammatory cytokines (such as IL-1, IL-6, and TNF_α_) upregulates the expression of tissue factor on monocytes and endothelial cells. This condition of impaired hemostatic function may be associated with a wide spectrum of clinical signs, which may vary from minor to severe bleeding [16]. In veterinary patients, DIC has been associated with different kinds of infections, including potential causes of AKI such as Leishmaniosis, Babesiosis, and Leptospirosis [10,18,19,20]. As several infectious agents may contribute to anemia, it is plausible that a cohort of AKI dogs with an elevated prevalence of infectious etiology would show a higher frequency of anemia.

The frequency of anemia increased significantly with the progression of the IRIS grade. Dogs of IRIS grades 3 to 5 showed a frequency of disorder between 64% and 83%. The higher frequency of anemia among dogs of later grades of AKI may be related to several factors such as reduction of circulating EPO, severity of azotemia, and oliguria [15]. As EPO is secreted by tubule-interstitial renal cells, it is plausible that AKI is responsible for a reduction in circulating EPO. The relation between EPO deficiency and increased frequency of anemia may be explained by the severity of renal injury. In human AKI patients, EPO levels have been shown to increase during the first 48 h, then to reduce progressively [21]. However, the role of EPO supplementation in AKI humans is still debated. A retrospective study by Park J and Colleagues in AKI patients showed that treatment with EPO was unable to lower transfusion requirements, and to improve renal recovery or survival [22]. Similar findings were reported in a recent multicentric study by Aoun M. and colleagues, in which the group of AKI human patients treated with EPO did not show a better outcome or a lower need for blood transfusion compared to the control group [15]. A key role in the pathophysiological mechanism of anemia may be played by indoxyl sulfate. This uremic toxin has been shown to promote anemia in CKD patients due to different mechanisms. Indoxyl sulfate promotes eryptosis by stimulating the entry of extracellular ionized calcium into erythrocytes, thus causing cell shrinkage and membrane scrambling. Indoxyl sulfate is also responsible for erythrocyte death due to the promotion of oxidative stress by NADPH oxidase activity-dependent and glutathione-independent mechanisms. Indoxyl sulfate has also been negatively associated with EPO expression and positively associated with EPO resistance and endothelial dysfunction [15]. This role of indoxyl sulfate was also confirmed by a study in CKD rats, in which the supplementation of AST-120 reduced serum indoxyl sulfate, and promoted EPO expression [23].

The frequency of anemic dogs was significantly higher than the frequency of non-anemic dogs for all the etiologies, with the exception of nephrotoxic AKI. Although the reason for the lower frequency of anemia in nephrotoxic AKI remains unknown, it is possible that the lower frequency of anemia may be secondary to a faster referral. It is known that the prognosis of dogs that ingested ethylene glycol is strictly dependent on the early starting of hemodialysis, so it is plausible that these dogs were referred to at the very beginning of the disease when anemia was still not present. On the other hand, the elevated frequency of anemia in AKI dogs is not surprising as anemia is a common finding in acute illness. In human medicine, anemia has been found in two-thirds of critically ill, hospitalized patients, showing that 97% of patients develop anemia by day eight of hospitalization [24]. In critically ill patients, the shortened life span of erythrocytes may be secondary to several causes such as hemolysis, gastrointestinal bleeding, oozing at injury sites, or blood samples [24]. Besides these mechanisms, our AKI dogs might also experience an impaired production of erythrocytes due to a combination of nutritional deficiency and inflammation. Anemia of inflammation has been identified as an inflammatory process responsible for impaired iron metabolism, proliferation of erythrocytes, and production and signaling of EPO [24].

According to therapeutical approach, the frequency of anemia was significantly higher in AKI dogs submitted to hemodialysis, compared to medically managed dogs. This difference may be secondary to the fact that hemodialysis is usually performed in dogs with a more severe degree of azotemia, or in case of failure of medical management. Therefore, the hemodialysis group might experience more severe conditions of azotemia and inflammation. Moreover, it is possible that in some dogs, hemodialysis was started as a rescue treatment after the failure of medical management. These dogs might be more severely uremic or uremic for a longer time, thus having more chances of developing anemia.

Non-survivor dogs were characterized by more severe forms of anemia, with a poorer degree of regeneration; however, the frequency of anemia did not differ significantly between survivors (74%) and non-survivors (64%). These findings seem to be in line with what was reported in human AKI patients. Although the underlying mechanism for the causal relation between AKI and anemia remains unclear, human AKI patients with anemia show a significantly higher mortality risk. The association between anemia and AKI may aggravate organ dysfunction and delay recovery [25].

The present study has several limitations. Firstly, the elevated frequency of dogs with AKI of unknown origin may represent a significant bias as it is possible that some dogs belonged to different categories. The second limitation is represented by the evaluation of anemia only at hospital admission with no information about its follow-up. The degree of regeneration of anemia has been assessed only as reticulocytes’ number and percentage. We cannot ignore that the group of dogs under medical management included good candidates for hemodialysis, which did not exert financial restraint. Finally, no information about long-term follow-up in terms of survival and recovery from AKI was available.

## 5. Conclusions

Anemia seems to be a very frequent disorder in AKI dogs, mostly characterized by normochromic, normocytic forms with poor regeneration rates. The frequency of anemia was elevated for all the etiologies, with increasing frequency with the progression of the IRIS grade. Although deceased dogs showed more severe and poorly regenerative forms, the frequency of anemia did not differ significantly between survivors and non-survivors. The elevated frequency of poor regenerative forms of anemia, also at early grades of AKI, should encourage to consider the use of erythrocyte stimulating agents in dogs with AKI

However, further studies are needed to investigate the effective role of an early correction of anemia on renal recovery and prognosis.

## Figures and Tables

**Table 1 vetsci-11-00212-t001:** Hemato-biochemistry parameters at hospital admission of AKI dogs in relation to AKI grade (a), etiology (b), therapeutical management (c), and outcome (d).

AKI Grade (a)	AKI (n = 120)	AKI 2 (n = 10)	AKI 3 (n = 22)	AKI 4 (n = 48)	AKI 5 (n = 40)	RR	*p*-Value
**RBC** (M/μL)	3.9 (1.5–9.4)	6.3 (1.5–9.1)	5.3 (2.3–9.4)	4.7 (1.6–8.7)	5.0 (1.8–8.3)	5.6–8.8	0.486
**HCT** (%)	31.8 ± 10.3	38.8 ± 15.3	33.0 ± 10.3	29.7 ± 8.7	32.0 ± 10.0	37.3–61.7	0.140
**HGB** (g/dL)	11.5 ± 3.7	13.0 ± 5.2	11.8 ± 3.8	10.9 ± 3.4	11.7 ± 3.6	13.1–20.5	0.540
**MCV** (fL)	63.2 (6.8–84.4)	64.5 (51.3–73.0)	63.4 (54.3–68.5)	63.0 (6.8–84.4)	63.3 (51.1–72.0)	61.6–73.5	0.975
**MCH** (pg)	23.1 (2.0–26.3)	21.2 (18.4–25.1)	22.9 (2.0–24.5)	23.2 (18.2–26.3)	23.2 (17.6–25.5)	21.2–25.9	0.155
**MCHC** (g/dL)	36.4 (15.6–40.9) *	33.8 (15.6–37.0) **	35.8 (31.9–39.9) *	37.0 (31.2–39.6) *	37.0 (33.8–40.9) *	32.0–37.9	0.001
**RDW** (%)	15.6 (0.4–32.2)	17.0 (0.4–25.5)	16.1 (13.8–32.2)	15.5 (12.2–32.2)	15.3 (12.7–26.7)	13.6–21.7	0.291
**%RET** (%)	0.4 (0.1–18.7)	0.7 (0.3–15.4)	0.3 (0.1–2.5)	0.6 (0.1–11.2)	0.3 (0.1–18.7)	0–10	0.077
**RET** (K/μL)	17.4 (3.2–236.8)	40.7 (15.4–97.7)	13.4 (7.7–97.7)	22.1 (3.8–236.8)	13.7 (3.2–60.7)	10–110	0.05
**Cr** (mg/dL)	8.3 (1.7–26.6)	2.1 (1.7–2.5) *	3.7 (2.6–5.0) *	8.0 (5.1–10.0) **	12.6 (10.6–26.6) ***	0.6–1.5	<0.0001
**Urea** (mg/dL)	303.1 ± 134.9	116.2 ± 41.5 *	204.1 ± 90.6 *	308.6 ± 100.7 **	397.9 + 121.4 ***	15–55	<0.0001
**TCa** (mg/dL)	10.1 ± 2.5	9.5 ± 0.8	9.9 ± 2.2	10.3 ± 3.1	10.0 ± 2.1	8.7–11.8	0.927
**P** (mg/dL)	13.2 ± 5.1 *	6.6 ± 2.8 **	9.0 ± 3.6 **	13.4 ± 4.4 *	15.9 ± 4.5 ***	2.5–5.0	<0.0001
**Na^+^** (mEq/L)	147 (117–172)	156 (140–166)	149 (117–161)	145 (123–171)	148 (122–172)	117–172	0.176
**K^+^** (mEq/L)	5.0 (2.5–11.0)	4.4 (3.8–8.8)	4.5 (2.5–7.3)	5.6 (2.7–9.0)	5.1 (3.4–11.0)	3.6–4.8	0.230
**Etiology (b)**	**IS**	**INF**	**NEP**	**OBS**	**UK**		***p*-value**
**RBC** (M/μL)	5.0 (1.9–9.1)	4.6 (1.6–7.3)	5.9 (3.4–8.7)	3.8 (1.5–8.1)	5.0 (1.8–9.4)		0.05
**HCT** (%)	31.3 ± 11.4	28.5 ± 7.9 *	39.0 ± 10.8 **	26.8 ± 16.8	32.9 ± 9.6		0.01
**HGB** (g/dL)	11.4 ± 4.2	10.4 ± 2.9 *	13.7 ± 3.9 **	9.1 ± 5.4	12.0 ± 3.6		0.02
**MCV** (fL)	62.0 (56.2–75.5)	61.5 (6.8–84.4) *	65.0 (62.1–73.3) **	65.3 (51.3–71.5)	63.3 (51.1–73.0)		0.02
**MCH** (pg)	23.1 (19.9–25.0)	22.8 (2.0–26.3)	23.4 (19.0–25.3)	22.2 (18.4–25.1)	23.2 (17.6–25.5)		0.567
**MCHC** (g/dL)	36.4 (33.1–38.3)	36.7 (31.2–40.9)	35.9 (27.8–38.1)	34.1 (15.6–35.8)	36.9 (31.9–39.4)		0.05
**RDW** (%)	15.8 (13.8–30.0)	15.9 (13.5–32.2)	15.1 (12.2–19.6)	16.9 (0.4–18.3)	15.4 (12.7–26.7)		0.486
**%RET** (%)	0.4 (0.1–10.4)	0.6 (0.1–11.2) *	0.2 (0.1–0.7) **	5.7 (0.4–15.4) *	0.4 (0.1–18.7)		0.006
**RET** (K/μL)	23.6 (5.5–200.3)	19.6 (4.4–185.5)	13.3 (3.5–44.2)	25.1 (15.4–236.8)	17.4 (3.2–97.7)		0.393
**Cr** (mg/dL)	8.1 (2.0–12.8)	6.8 (2.3–17.0) *	7.0 (1.7–26.6)	2.5 (1.7–8.4) *	10.2 (1.9–25.6) **		0.002
**Urea** (mg/dL)	319.4 ± 117.5	297.4 ± 139.7	262.6 ± 192.2	166.6 ± 98.3	327.8 ± 112.1		0.07
**TCa** (mg/dL)	9.3 ± 1.8	10.7 ± 2.4	9.8 ± 2.1	9.3 ± 1.2	10.3 ± 2.4		0.380
**P** (mg/dL)	13.4 ± 4.0	12.9 ± 5.1	10.6 ± 6.2	7.9 ± 3.7	14.5 ± 4.6		0.03
**Na^+^** (mEq/L)	145.5 (136.0–160.0)	145.0 (122.0–159.0)	145.6 (134.0–166.0)	144.0 (139.0–148.0)	149.0 (117.0–172.0)		0.417
**K^+^** (mEq/L)	4.6 (3.3–8.8)	4.9 (2.7–9.0)	4.7 (3.4–11.0)	4.8 (3.8–6.1)	5.5 (2.5–8.4)		0.319
**Therapeutical management (c)**	**Medical**		**Hemodialysis**				***p*-value**
**RBC** (M/μL)	5.3 (1.5–9.4)		4.8 (1.8–8.7)				0.464
**HCT** (%)	33.0 ± 11.7		31.0 ± 9.1				0.285
**HGB** (g/dL)	11.9 ± 4.4		11.3 ± 3.2				0.374
**MCV** (fL)	32.6 (8.1–58.2)		30.3 (10.7–55.7)				0.321
**MCH** (pg)	23.2 (17.6–26.3)		23.1 (2.0–25.8)				0.358
**MCHC** (g/dL)	36.1 (15.6–39.1)		36.7 (27.8–40.9)				0.09
**RDW** (%)	15.6 (0.4–32.2)		15.6 (12.2–26.0)				0.737
**%RET** (%)	0.5 (0.1–15.4)		0.4 (0.1–18.7)				0.572
**RET** (K/μL)	17.4 (3.8–200.3)		17.4 (3.2–236.8)				0.416
**Cr** (mg/dL)	6.2 (1.7–20.6)		9.0 (1.7–26.6)				0.01
**Urea** (mg/dL)	273.7 ± 132.3		324.1 ± 133.7				0.04
**TCa** (mg/dL)	10.0 ± 2.1		10.3 ± 2.4				0.498
**P** (mg/dL)	12.6 ± 5.2		13.6 ± 4.9				0.340
**Na^+^** (mEq/L)	149.0 (117.0–172.0)		145.2 (122.0–166.0)				0.07
**K^+^** (mEq/L)	5.3 (2.5–8.8)		4.9 (3.2–11.0)				0.401
**Outcome (d)**	**Survivors**		**Non-survivors**				***p*-value**
**RBC** (M/μL)	4.8 (1.5–8.7)		4.9 (1.6–9.4)				0.612
**HCT** (%)	32.1 ± 9.5		31.4 ± 11.2				0.695
**HGB** (g/dL)	11.5 ± 3.3		11.5 ± 4.2				0.982
**MCV** (fL)	63.0 (6.8–73.0)		63.6 (56.0–84.4)				0.335
**MCH** (pg)	23.0 (2.0–25.4)		23.2 (19.9–26.3)				0.01
**MCHC** (g/dL)	36.2 (15.6–39.6)		37.0 (31.2–40.9)				0.111
**RDW** (%)	15.7 (0.4–26.7)		15.3 (12.2–32.2)				0.242
**%RET** (%)	0.4 (0.1–18.7)		0.5 (0.1–11.2)				0.703
**RET** (K/μL)	17.3 (3.5–236.8)		17.4 (3.2–200.3)				0.687
**Cr** (mg/dL)	7.2 (1.7–22.6)		9.5 (2.0–26.6)				0.02
**Urea** (mg/dL)	279.7 ± 125.6		331.8 ± 141.3				0.03
**TCa** (mg/dL)	10.7 ± 2.3		9.6 ± 2.2				0.01
**P** (mg/dL)	11.5 ± 4.5		15.2 ± 5.0				0.0002
**Na^+^** (mEq/L)	149.0 (134.0–172.0)		144.0 (117.0–171.0)				0.03
**K^+^** (mEq/L)	4.9 (3.3–9.0)		5.1 (2.5–11.0)				0.420

IS: ischemic/inflammatory; INF: infective; NEP: nephrotoxic; OBS: obstructive; UK: unknown; Min: minimum; Max: maximum; RR: reference range of the laboratory; Cr: serum creatinine (mg/dL); TCa: serum total calcium (mg/dL); P: serum phosphate (mg/dL); Na^+^: serum sodium (mEq/L); K^+^: serum potassium (mEq/L). For non-parametric variables, Mann–Whitney test was used to compare median values between two groups, while Kruskal–Wallis test and Dunn’s test were used when more than two groups were present. For parametric variables, unpaired *t*-test was used to compare mean values between two groups, while One-way ANOVA and Tukey’s test were used when more than two groups were present. Statistical significance was set at *p* < 0.05. Statistically significant difference among groups was indicated by *, ** and ***.

**Table 2 vetsci-11-00212-t002:** Chi-squared comparison of the frequency of anemia, degree of anemia, and regeneration rate according to AKI grade, etiology, therapeutical management, and survival.

Variables	Anemia (%)		Severity (%)		Degree of Regeneration (%)		
	Yes	No	Mild to Moderate	Severe to Very Severe	None	Mild	Moderate
**AKI grade** AKI 2 AKI 3 AKI 4 AKI 5	40% 64% 83% 70% *p* < 0.0001	60% 36% 17% 30%	75% 86% 83% 85% *p* = 0.163	25% 14% 17% 15%	100% 85% 86% 100% *p* < 0.0001	- 15% 7% -	- - 7% -
**Etiology** Ischemic/Inflammatory Infectious Nephrotoxic Obstructive Unknown	82% 81% 43% 80% 70% *p* < 0.0001	18% 19% 57% 20% 30%	72% 80% 100% 50% 94% *p* < 0.0001	28% 20% - 50% 6%	79% 94% 100% 75% 94% *p* < 0.0001	14% 3% - - 6%	7% 3% - 25% -
**Therapeutical approach** Medical management Hemodialysis	64% 77% *p* = 0.04	36% 23%	84% 83% *p* = 0.848	16% 17%	88% 91% *p* = 0.141	6% 8%	6% 1%
**Outcome** Survived Deceased	74% 64% *p* = 0.433	23% 36%	92% 73% *p* = 0.0004	8% 27%	88% 92% *p* = 0.07	10% 3%	2% 5%

## Data Availability

Data are available in the database of the Veterinary Teaching Hospital.

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
