# Peer review of "Anemia in Dogs with Acute Kidney Injury"

_vetsci, 2024, doi:10.3390/vetsci11050212_

Round 1
Reviewer 1 Report
Comments and Suggestions for Authors
Line 23: infectious INF) correct (INF)
Line 30: references to classify reticulocyte guideline counts in canine are variable. The cases described do not show marked regeneration, and tend to be equivocal or mild (again, depending on classification).
Line 50: Is AKI azotemic in any IRIS stage (uremia)? Later no dogs were in stage IRIS 1 all your results show IRIS 2 and higher.
Line 54: AKi may potentially be caused by urinary tract rupture
Line 197: I think there is no need to write Akita-Inu, Akita will suffice since inu means dog in japanese. Chnge it if you agree.
Line 228: hyperchromic anemia worth wile to be defined and discussed, you provide a lot of relevant informastion in. the discussion.
Author Response
Line 23: infectious INF) correct (INF)
Line 23 has been corrected according to the Reviewer’s comment
Line 30: references to classify reticulocyte guideline counts in canine are variable. The cases described do not show marked regeneration, and tend to be equivocal or mild (again, depending on classification).
We agree with the Reviewer that there is a significant variability in the references used for classification of regeneration rate. We opted to follow the guidelines of Tvedten H et al 2010, in order to be consistent with the classification we used in a previous study of anemia in CKD dogs.
Line 50: Is AKI azotemic in any IRIS stage (uremia)? Later no dogs were in stage IRIS 1 all your results show IRIS 2 and higher.
The Reviewer’s comment is correct, as uremia is a consequence of more severe forms of AKI. Unfortunately, the usual parameters of renal function are inadequate to diagnose early grades of AKI (such as AKI grade 1). These grades may be diagnosed only by the use of early markers of glomerular and tubular damage, which are not currently available in clinical setting.
Line 54: AKi may potentially be caused by urinary tract rupture
We agree with the Reviewer’s comment, so we added rupture of the urinary tract among the possible cause of AKI. Please refer to line 54
Line 197: I think there is no need to write Akita-Inu, Akita will suffice since inu means dog in japanese. Chnge it if you agree.
In agreement with the Reviewer’s comment we opted to change Akita-Inu to Akita
Line 228: hyperchromic anemia worth wile to be defined and discussed, you provide a lot of relevant informastion in. the discussion.
In agreement with the Reviewer’s comment, we opted to discuss the elevation of the MCHC in the discussion section. Please refer to line 293-294
Reviewer 2 Report
Comments and Suggestions for Authors
Review of the article: Anemia in dogs with acute kidney injury.
The authors analysed associations between acute kidney injury and anemia in dogs, including association between severity and frequency of anemia and stage of AKI. The study is well designed, and the article well written. However, I have some specific comments which should be responded by the authors.
Specific comments:
The whole text: The authors use the word uremia, and define its severity using serum creatinine concentration (lines 134-135). However, uremia is associated with clinical state, and dogs with uremia have various clinical signs. Yet, the authors of this work did not present any spectrum of clinical signs in animals included to the study. There is only comparison of urea and creatinine concentration. Thus, the authors should use the word azotemia in the whole text, as they treat it as laboratory (not clinical) word.
Lines 61-63: This statement is present in the discussion section of the cited reference No. 4. However, it is citation of the statement from another work (https://doi.org/10.3109/08860229409044854), and that article should be cited here.
Lines 123-125: It was mentioned above (line 106) that dogs with diseases affecting CBC were excluded from the study. However, all these infectious diseases may affect CBC.
Lines 153-156: Please add the reference for these values (MCV and MCHC) - these are not from the reference cited here i.e. reference No. 14.
Line 206: “infective”. What were the infections? How were they recognized? The authors mentioned earlier babesiosis and other diseases. However, it was shown in a previous study that there was no association between anemia and azotemia in dogs infected with Babesia (Zygner and Wedrychowicz. 2009, Influence of anaemia on azotaemia in dogs infected with Babesia canis in Poland).
Table 1.:
-What do the asterisks mean in the table?
-Why the number of dogs in AKI (n=121) is 121? It was mentioned earlier that 120 dogs were included to the study.
-Column RR: Please add the reference for the reference intervals.
-Reference interval for hematocrit: Reference range is between 37.3 and 61.7. It means that dogs with hct lower than 37.3 % (not 37 %, as it was mentioned in materials and methods) should be considered as anemic.
Discussion section: Good discussion. However, lack of discussion of some results e.g. statistically significant difference in sodium concentration between survivors and non-survivors. This difference may result from SIADH development, which is prevalent in hospitalized humans, and was described in dogs infected with Babesia (https://doi.org/10.2478/jvetres-2019-0045).
Author Response
The whole text: The authors use the word uremia, and define its severity using serum creatinine concentration (lines 134-135). However, uremia is associated with clinical state, and dogs with uremia have various clinical signs. Yet, the authors of this work did not present any spectrum of clinical signs in animals included to the study. There is only comparison of urea and creatinine concentration. Thus, the authors should use the word azotemia in the whole text, as they treat it as laboratory (not clinical) word.
In agreement with the Reviewer’s comment, we opted to change the term uremia to azotemia throughout the manuscript. Unfortunately, clinical signs of uremia could not be included in the manuscript, due to space issues.
Lines 61-63: This statement is present in the discussion section of the cited reference No. 4. However, it is citation of the statement from another work (https://doi.org/10.3109/08860229409044854), and that article should be cited here.
The Reviewer is right, as lines 61-63 refer to Hales et al 1994. However, we opted to cite here also Powell-Tuck et al 2016, as we added information about the controversial relationship between anemia and AKI progression. Please refer to lines 61-63
Lines 123-125: It was mentioned above (line 106) that dogs with diseases affecting CBC were excluded from the study. However, all these infectious diseases may affect CBC.
We agree with the Reviewer that all the overmentioned infectious disease can affect significantly CBC, however, these infectious represent a significant part of the common causes of AKI in dogs. Therefore we opted to include them in the study population.
Lines 153-156: Please add the reference for these values (MCV and MCHC) - these are not from the reference cited here i.e. reference No. 14.
Reference 14 refers to the classification of the severity and regeneration of anemia. For MCV and MCHC we used the reference ranges of our laboratory of clinical pathology.
Line 206: “infective”. What were the infections? How were they recognized? The authors mentioned earlier babesiosis and other diseases. However, it was shown in a previous study that there was no association between anemia and azotemia in dogs infected with Babesia (Zygner and Wedrychowicz. 2009, Influence of anaemia on azotaemia in dogs infected with Babesia canis in Poland).
The infective group included dogs, in which the cause of AKI was secondary to infectious disease. Among these dogs we included dogs with acute bacterial pyelonephritis (E.Coli; Staphylococcus; …), and acute forms of Leptospirosis, Babesiosi, Ehrlichiosis, and Leishmaniosi). We did non indagate the relationship between each infectious agent and anemia. We only considered these infectious agents as the cause of the AKI.
Table 1.:
-What do the asterisks mean in the table?
The asterisks indicate the presence of statistical significance at post-hoc test
-Why the number of dogs in AKI (n=121) is 121? It was mentioned earlier that 120 dogs were included to the study.
There was a typing error In Table 1. We amended the table by indicating the right number.
-Column RR: Please add the reference for the reference intervals.
The reference ranges for the different variables refer to the ranges of the laboratory of clinical pathology of our hospital. We added this information in Table 1.
-Reference interval for hematocrit: Reference range is between 37.3 and 61.7. It means that dogs with hct lower than 37.3 % (not 37 %, as it was mentioned in materials and methods) should be considered as anemic.
Although the reference range of our laboratory set the lower level of HCT at 37.3%, we opted to consider dogs with reduced HCT only if their HCT was below 30%, as stated in the materials and methods section. This cut off value was chosen in order to be consistent with the cut off we used in our previous paper about anemia in CKD dogs (Lippi et al 2021)
Discussion section: Good discussion. However, lack of discussion of some results e.g. statistically significant difference in sodium concentration between survivors and non-survivors. This difference may result from SIADH development, which is prevalent in hospitalized humans, and was described in dogs infected with Babesia (https://doi.org/10.2478/jvetres-2019-0045).
We agree with the Reviewer that some results (such as the sodium) should be extremely interesting to be discussed, especially in relation to survival. The topic of SIADH is also of great interest. However, the discussion section was mainly focused on the interactions between anemia and AKI. These interactions and mechanisms are so numerous, that we think that adding other topics may cause a significant extension of the discussion
Reviewer 3 Report
Comments and Suggestions for Authors
Simple Summary –
Line 11 – reword “very elevated frequency” – consider common
Line 12 – reword to dogs with AKI not AKI dogs
Line 12 – showed “a” mild to moderate…
Abstract
Line 18 – is it a disorder or complication?
Line 19 – concise wording to “…retrospectively evaluate frequency, severity and regeneration…”
Line 25 – and survival to discharge…
Line 25 – consider wording as “anemia was defined as” instead of “aki dogs”
Line 29 – how were these “mild, moderate” etc chosen
Line 36 – be consistent in terminology – progression is used but other places it’s specific when talking about IRIS
Line 51 – replace “were” with are
Line 54 – citation
Line 57 -citation
Line 57 – causes associated with mortality – unlikely to be important in the intro
Line 63 – “is” postulated
Line 68-75 – this is saying that the anemia is contributing to the AKI – not that AKI is causing the anemia
- This entire paragraph is anemia causing AKI and not that AKI is associated with anemia
Line 80 – yes CKD is associated with anemia – however, it doesn’t say why that’s an important finding
Line 83 – this makes it sound like this question as already been answered – the incidence has been reported so what is different about the current, present manuscript
Line 96 – not enough information – does this mean there was prior bloodwork and creat of 0.3 was counted?
The case selection does not have enough information – did they have prior bloodwork? Was this at presentation or any time during hospitalization?
Therapeutic management – seems like the dialysis would be a confounding factor since not all cases that qualified received it
Line 224 – there is not enough information – was the CBC at the same time as diagnosis of the AKI -did they have other confounding factors
There is not enough information on timing – one time diagnosis of anemia? When is the timing of reticulocytes – can this be pre-regenerative?
What other treatments?
Discussion – this is not explaining why the findings of your study are different than previous study
Overall – there does not seem to be enough information about study criteria and the overall conclusions
Comments on the Quality of English LanguageMinor edits required
Author Response
Simple Summary –
Line 11 – reword “very elevated frequency” – consider common
In agreement with the Reviewer’s comment, we opted to change line 11, as indicated. Please refer to line 11.
Line 12 – reword to dogs with AKI not AKI dogs
In agreement with the Reviewer’s comment, we rephrased the text.
Line 12 – showed “a” mild to moderate…
In agreement with the Reviewer’s comment we rephrased the text
Abstract
Line 18 – is it a disorder or complication?
In agreement with the Reviewer’s comment we opted to change the term “disorder” to “complication”. Please refer to the text
Line 19 – concise wording to “…retrospectively evaluate frequency, severity and regeneration…”
In agreement with the Reviewer’s comment, line 19 has been rephrased.
Line 25 – and survival to discharge…
In agreement with the Reviewer’s comment line 25 has been rephrased.
Line 25 – consider wording as “anemia was defined as” instead of “aki dogs”
In agreement with the Reviewer’s comment line 25 has been rephrased.
Line 29 – how were these “mild, moderate” etc chosen
The degree of anemia was chosen on the basis of what reported by Tvedten et al 2010
Line 36 – be consistent in terminology – progression is used but other places it’s specific when talking about IRIS
In agreement with the Reviewer’s comment with rephrased line 36
Line 51 – replace “were” with are
In agreement with the Reviewer’s comment “were” was replaced by “are”
Line 54 – citation
The citation of line 54 is reported below (Legatti et al 2018)
Line 57 -citation
The citation of line 54 is the same reported below (Legatti et al 2018)
Line 57 – causes associated with mortality – unlikely to be important in the intro
In agreement with the Reviewer’s comment we opted to remove this line from the introduction.
Line 63 – “is” postulated
In agreement with the Reviewer’s comment the word “was” was changed to “is”
Line 68-75 – this is saying that the anemia is contributing to the AKI – not that AKI is causing the anemia
- This entire paragraph is anemia causing AKI and not that AKI is associated with anemia
In the first lines of the paragraph we reported the fact that anemia may be more frequent in AKI population compared to non AKI population, as reported by Nishimoto et al 2020. In the second part of the paragraph we reported the possible role of pre-existing anemia in promoting AKI
Line 80 – yes CKD is associated with anemia – however, it doesn’t say why that’s an important finding
In agreement with the Reviewer’s comment, we added the main reported mechanisms associated with progression of the IRIS stage in CKD patients with anemia. Please refer to line 80
Line 83 – this makes it sound like this question as already been answered – the incidence has been reported so what is different about the current, present manuscript
In the study of Rimer and Colleagues, anemia was considered as part of other clinical and hematological parameters in a group of AKI dogs. In our study we tried to do a step forward, by investigating also its degree of severity, and regeneration rate in relation to IRIS grade, etiology, therapy, and outcome
Line 96 – not enough information – does this mean there was prior bloodwork and creat of 0.3 was counted?
The case selection does not have enough information – did they have prior bloodwork? Was this at presentation or any time during hospitalization?
We thank the Reviewer for the comment. Dogs were included in the study and classified on the basis of the serum creatinine they showed at hospital admission. For this reason, IRIS grade 1 could not be included, as we should have performed early markers of AKI, or serial measures of serum creatinine. The diagnosis of AKI based not only on the finding of abnormal serum creatinine, but also on the concomitant presence of a history suggestive of acute renal failure, and absence of ultrasound findings of CKD. As suggested by the Reviewer, we tried to clarify these inclusion criteria at line 96. Please refer to the Materials and methods section.
Therapeutic management – seems like the dialysis would be a confounding factor since not all cases that qualified received it
In agreement with the Reviewer’s comment, we opted to underline this point among the main limitations of the study. Please refer to the Discussion section
Line 224 – there is not enough information – was the CBC at the same time as diagnosis of the AKI -did they have other confounding factors
As stated in the Materials and Methods section, all the hematological parameters of the study, as well as the subsequent classifications of anemia, referred to hospital admission. In order to clarify this point, lines 102 to 104 have been modified.
There is not enough information on timing – one time diagnosis of anemia? When is the timing of reticulocytes – can this be pre-regenerative?
What other treatments?
We agree with the Reviewer that, as the hemato-biochemical parameters, that we included in the study, referred to hospital admission, we have no information concerning timing of anemia, and/or regenerative response. In order to avoid possible bias, we opted to exclude all dogs which had already received treatments able to significantly affect anemia (alpha-darbepoetin/ blood transfusion)
Discussion – this is not explaining why the findings of your study are different than previous study
In order to clarify what the Reviewer already reported in the comment to line 83, we opted to insert a comment into the conclusion session. Please refer to Conclusion
Overall – there does not seem to be enough information about study criteria and the overall conclusions
Please refer to the above comments
Round 2
Reviewer 3 Report
Comments and Suggestions for Authors
It does not appear that the suggestions/concerns were addressed.
Comments on the Quality of English LanguageEdits are required
Author Response

(The authors gave the same response as above.)
